# Bacterial Lipopolysaccharide Induces PD-L1 Expression and an Invasive Phenotype of Oral Squamous Cell Carcinoma Cells

**DOI:** 10.3390/cancers16020343

**Published:** 2024-01-13

**Authors:** Yuji Omori, Kazuma Noguchi, Mizuha Kitamura, Yuna Makihara, Takayuki Omae, Soutaro Hanawa, Kyohei Yoshikawa, Kazuki Takaoka, Hiromitsu Kishimoto

**Affiliations:** Departments of Oral and Maxillofacial Surgery, School of Medicine, Hyogo Medical University, Mukogawa-cho1-1, Nishinomiya 663-8501, Japan; yu-oomori@hyo-med.ac.jp (Y.O.); mi-kitamura@hyo-med.ac.jp (M.K.); yu-makihara@hyo-med.ac.jp (Y.M.); ta-oomae@hyo-med.ac.jp (T.O.); so-hanawa@hyo-med.ac.jp (S.H.); kyo331@hyo-med.ac.jp (K.Y.); ktaka@hyo-med.ac.jp (K.T.); kisihiro@hyo-med.ac.jp (H.K.)

**Keywords:** PD-L1, partial EMT, LPS, TLR4, *Porphyromonas gingivalis*

## Abstract

**Simple Summary:**

There are many aspects of oral cancer invasion and metastasis that remain to be elucidated. It has been suggested that chronic inflammation caused by bacteria may be a cause of carcinogenesis, invasion, and metastasis. PD-L1 expression is associated with lymph node metastasis and poor prognosis in several cancers. Lipopolysaccharide (LPS) from *Porphyromonas gingivalis*, which is known to be the main cause of bacterial periodontitis, induced PD-L1 expression and partial epithelial–mesenchymal transition (pEMT) expression in oral cancer cells. Moreover, we identified PD-L1 expression within the exosomes of oral cancer cells. Exosomes may function as carriers of PD-L1.

**Abstract:**

Background: Expression of programmed death ligand-1 (PD-L1) is related to the prognosis of many solid malignancies, including oral squamous cell carcinoma (OSCC), but the mechanism of PD-L1 induction remains obscure. In this study, we examined the expression of PD-L1 and partial epithelial–mesenchymal transition (pEMT) induced by bacterial lipopolysaccharide (LPS) in OSCC. Methods: The expression of Toll-like receptor 4 (TLR4) recognizing LPS in OSCC cell lines was analyzed. Moreover, the induction of PD-L1 expression by *Porphyromonas gingivalis* (*P.g*) or *Escherichia coli* (*E. coli*) LPS and EMT was analyzed by western blotting and RT-PCR. Morphology, proliferation, migration, and invasion capacities were examined upon addition of LPS. PD-L1 within EXOs was examined. Results: PD-L1 expression and pEMT induced by LPS of *P.g* or *E. coli* in TLR4-expressing OSCC cell lines were observed. Addition of LPS did not change migration, proliferation, or cell morphology, but increased invasive ability. Moreover, higher expression of PD-L1 was observed in OSCC EXOs with LPS. Conclusion: Oral bacterial LPS is involved in enhanced invasive potential in OSCC cells, causing PD-L1 expression and induction of pEMT. The enhancement of PD-L1 expression after addition of LPS may be mediated by EXOs.

## 1. Introduction

Oral cancer, represented as oral squamous cell carcinoma (OSCC) in more than 90% of all oral cavity malignancies, is an important cause of cancer mortality worldwide, with an estimated 177,000 deaths every year [1]. It is considered to be an aggressive cancer, with a 5-year overall survival rate of approximately 50% that reduces to less than 30% at advanced stages [2]. Tumor extension, lymph node metastasis, high rates of locoregional recurrence, and development of second primary tumors are the leading causes of death in OSCC patients [3].

Primary tumor depth of invasion is currently the most reliable measure to estimate risk of occult nodal metastasis in OSCC [4]. Primary tumor depth of invasion, tumor budding, and tumor invasion status are related to epithelial-to-mesenchymal transition (EMT). EMT is the process by which epithelial cells lose their cell adhesion function and polarity and gain migration and invasive capacities to generate mesenchymal cell types [5]. This phenomenon is considered reversible, and the opposite phenomenon, mesenchymal-to-epithelial transition (MET), is also known to occur [6]. EMT is generally triggered by external signaling such as TGF-β, Notch ligands, Wnt, and hypoxia [7,8,9]. Recently, it has been reported that periodontal bacteria induce EMT [10]. A pooled meta-analysis of well-known EMT transcription factors (EMT–TFs) in head and neck squamous cell carcinoma (HNSCC) showed a strong correlation between the expression of EMT–TFs and poor overall survival [11]. The overexpression of TFs *TWIST1*, *SNAI1*, *SNAI2*, and *ZEB1* in particular showed a significant association with poor overall survival among HNSCC patients [12]. Recently, Pastushenko et al. reported that cancer cells not only undergo such changes and partial EMT (pEMT) based on their plasticity, but also undergo these in other situations such as contact with the stroma and hypoxia. Moreover, it has become clear that pEMT may be associated with resistance to radiation and chemotherapy, as well as immune escape from the host [13]. Besides having a heightened invasive capacity to circulate to distant metastatic sites, pEMT cells may also gain stemness properties [14,15,16,17].

Anti-PD-1 antibodies pembrolizumab (PEM) and nivolumab are currently used in the treatment of patients with relapsed HNSCC [18,19,20]. Tumor-expressed PD-L1 has been implicated in immune escape and malignant transformation, and increased PD-L1 expression is a predictor of poor prognosis [21]. Factors of enhanced PD-L1, including the EGF–EGFR and JAK–STAT–IL6R signaling pathways, radiation, and chemotherapy, and recently, chronic inflammation, are also known to induce PD-L1. 

In this report, we examined whether *Porphyromonas gingivalis* (*P.g*) LPS, which commonly causes chronic periodontitis and has been linked to OSCC, induces an increase in PD-L1 and pEMT. Furthermore, pEMT in OSCC is regulated by the extracellular matrix and exosomes (EXOs). EXOs from OSCC were extracted, and the expression of PD-L1 in EXOs was examined. 

## 2. Materials and Methods

### 2.1. Culture of OSCC Cell Lines

Gingival cancer cell lines HCMSqCC010 [22] and SCCKN [23] were established in our department. Ca9-22, SAS, SCC25, HSC-3, and HSC3-M were purchased from the Japan Research Resources Cell Bank (Osaka, Japan). OSC19 and OSC20 were kindly provided by Prof. Kawashiri, Kanazawa University, Japan. UPCI-SCC090, an HPV-positive tongue cancer cell line, was also provided by the University of Pittsburgh, Cancer Institute. HCMSqCC010 and HCMSqCC020 [22] were incubated in F-medium [24], and the other cell lines were incubated in Dulbecco’s Modified Eagle’s Medium (DMEM) with 10% FBS at 37 °C and 5% CO_2_.

### 2.2. Western Blotting

After cell culture, cells were washed with PBS (Mg^2+^ and Ca^2+^ free) and centrifuged; RIPA buffer (cat. no. sc-24948; Santa Cruz, Inc., Dallas, TX, USA) was added, and cells were incubated at 4 °C for 60 min, then centrifuged at 12,000× *g* for 20 min at 4 °C. The supernatant was used as the total cell lysate, and proteins were extracted. Protein concentrations were measured using the Bradford assay [25]. Western blotting (WB) was performed as previously described [26]. Signals were detected by chemiluminescence using a Pierce SuperSignal western blotting kit (TherE.colimo Fisher Scientific, Waltham, MA, USA). The primary and secondary antibodies are shown in Appendix A.

### 2.3. Reverse Transcription PCR

RT-PCR was performed as previously described [27]. RNA was extracted using TRIzol (Invitrogen, Carlsbad, CA, USA), and reverse transcription PCR was performed using Prime SCRIPT RT-PCR kit (Takara Bio, Kusatsu, Japan) according to the manufacturer’s instructions. The reagents in the kit were adjusted and then denatured and annealed at 65 °C for 5 min. Then, the reagents were further adjusted, and the mixture was denatured and annealed at 30 °C for 10 min, 42 °C for 30 min, and overnight at 4 °C. The next day, the primers to be examined were prepared, and the reaction was repeated for 35 cycles of 10 s at 98 °C, 10 s at 60 °C, and 1 min at 68 °C. The sequences of the primers used are shown in Appendix A, and *GAPDH* was amplified as a control.

### 2.4. Cell Proliferation Assay

Ca9-22, OSC20, OSC19, and SAS cells (5000 cells/well) were plated in 96-well plates with or without 1 µg/mL *P.g* LPS (Sigma–Aldrich, St. Louis, MO, USA) or *E. coli* LPS (Sigma–Aldrich) as a positive control [28]. A Cell Counting Kit-8 (Dojindo Molecular Technologies, Kumamoto, Japan) was used to analyze cell counts after 6, 30, and 48 h of treatment. After incubation at 37 °C with the reagent, optical density was read at 450 nm using a Benchmark Plus microplate reader (BIO-RAD, Hercules, CA, USA).

### 2.5. Scratch Assay

Confluent monolayers were prepared by plating Ca9-22, OSC20, OSC19, and SAS cells in 60 mm dishes. The cell monolayer was then scraped off in a straight line with a pipette tip. Cells were washed once, and 1 µg/mL and 5 µg/mL *P.g* or *E. coli* LPS were added, with the reference points filled in by phase contrast microscopy. The cells were incubated in an incubator at 37 °C for 24 h, and the distance of each scratch closure was measured [29].

### 2.6. Invasion Assay

Invasion assays were performed using a CytoSelectTM 24-well cell invasion assay kit according to the manufacturer’s instructions (Cell Biolabs, San Diego, CA, USA) [30] (Figure 1). Ca9-22, OSC20, OSC19, and SAS cell suspensions at 1.0 × 10^6^ cells/mL in serum-free media were prepared. Then, 500 µL media containing 10% fetal bovine serum and 1 µg/mL *P.g* or *E. coli* LPS were added to the lower well of the invasion plate. Next, 300 µL cell suspension was added to the inside of each insert, and plates were incubated for 24 h at 37 °C in a 5% CO_2_ atmosphere. The media were then carefully aspirated from the inside of the insert. The ends of 2–3 cotton-tipped swabs were soaked with water and flattened by pressing against a clean hard surface, and the interiors of the inserts were gently swabbed to remove non-invasive cells. Each insert was then transferred to a clean well containing 400 µL Giemsa and incubated at room temperature for 10 min. The stained inserts were gently washed several times in a beaker of water and allowed to air dry. Invasive cells were counted with a light microscope under a high magnification, with three individual fields per insert.

### 2.7. Evaluation of Cell Morphology

Ca9-22, OSC20, OSC19, and SAS cells were treated with 10 ng/mL TGF-β and 1 µg/mL *P.g* or *E. coli* LPS and cultured for 48 h. We evaluated the transformation of each cell morphology by phase contrast microscopy (TE300, Nikon, Tokyo, Japan).

### 2.8. Evaluation of Plasticity of Cellular Changes

To evaluate plasticity, 10 ng/mL TGF-β and 1 µg/mL *E. coli* or *P.g* LPS were added to Ca9-22, OSC20, OSC19, and SAS cells and cultured for 48 h, followed by replacement of the media with LPS-free medium and culturing for a further 48 h. PD-L1 expression was then evaluated by WB.

### 2.9. Isolation and Characterization of EXOs

EXOs were isolated from the culture medium using the ExoQuick™ Exosome Precipitation kit (System Bioscience, Palo Alto, CA, USA) according to the manufacturer’s instructions. Briefly, EXOs were isolated from OSCC-conditioned medium by ultracentrifugation after 48 h of incubation in fresh medium containing FBS with EXOs removed, and the supernatant was centrifuged at 2000× *g* for 10 min at room temperature [22]. EXO-containing pellets were analyzed by WB, and the EXO markers TSG101 and CD63 were identified. Transmission electron microscopy (TEM) was also performed. Ca9-22, OSC20, OSC19, and SAS cells were incubated with 1 µg/mL *P.g* LPS for 48 h, and EXOs were extracted to evaluate PD-L1 expression in EXOs compared with EXOs in OSCC cells without LPS addition.

### 2.10. Statistical Analysis

For all data sets, the Mann–Whitney U test was used to evaluate significant differences, with a *p*-value of <0.05 being considered statistically significant (* *p* < 0.05, ** *p* < 0.01).

## 3. Results

### 3.1. TLR4 Expression in OSCC Cells

TLR4 has been identified as a molecule that recognizes LPS, a bacterial component of Gram-negative bacteria. Moreover TLR4, which is expressed in non-small cell lung cancer, is known to induce PD-L1 expression upon stimulation with *E. coli* LPS [31]. TLR4 expression was observed in 6 of 12 OSCC cell lines by WB (Figure 2). Among the TLR4-expressing OSCCs, HSQ89 and UPCISCC090 were excluded from this experiment because HSQ89 cells are from maxillary sinus cancer and UPCISCC090 cells are HPV-positive OSCCs. OSCC cell lines with positive TLR4 expression, OSC20, OSC19, and SAS, which originated from tongue cancer, and OSCC cells with negative TLR4 expression, Ca9-22, which were from gingival carcinoma, were examined.

### 3.2. Evaluation of EMT and pEMT of OSCC Cells following LPS Addition

When *P.g* or *E. coli* LPS was added to the above four OSCC cell lines and cultured, TLR4-positive OSCC cells showed decreased expression of E-cadherin and enhanced expression of N-cadherin by WB with the addition of LPS. Moreover, the addition of LPS enhanced the expression of LAMB3, a marker of pEMT (Figure 3). Previous studies have shown that the addition of TGF-β induces EMT [32]. RT-PCR showed that the addition of TGF-β enhanced the expression of the gelatinase group of MMPs or EMT–TF system, such as *MMP2*, *MMP9*, *SNAI*, and *ZEB* (Figure 4A). Similar upregulation was also observed when *P.g* LPS was added (Figure 4B).

### 3.3. Evaluation of Cell Proliferative Ability, Migration Ability, Invasion Ability, and Cell Morphology of OSCC Cells with LPS Addition

When *P.g* or *E.coli* LPS was added to OSCC cells, their proliferative and migration abilities were unchanged (Figure 5 and Figure 6). The spindle shape of the four OSCC cell lines was observed by adding TGF-β or *E. coli* LPS but not by the addition of *P.g* LPS (Figure 7). Cell invasion was increased when *P.g* or *E. coli* LPS was added (Figure 8A,B). OSC20 showed increased expression of *MMP2* and *MMP9* with the addition of LPS. Similarly, OSC20 tended to be more invasive than the other three OSCC cell lines. The difference in the invasive ability induced by *P.g* and *E. coli* was not definitive.

### 3.4. PD-L1 Expression Reversibility of OSCC Cells

Stimulation with LPS induces PD-L1 expression [31]. Therefore, we examined PD-L1 expression under *P.g* or *E. coli* LPS stimulation in TLR4-expressing OSC20, OSC19, and SAS cells. The expression of PD-L1 was increased by stimulation with LPS (Figure 9A). Moreover, in the cellular change reversibility experiment, a decrease in PD-L1 expression was observed in OSCC cells cultured with *P.g* LPS and then additionally cultured in medium without LPS for 2 days (Figure 9B).

### 3.5. Extraction and Analysis of Exosomes from OSCC Cells

EXOs were extracted using the above method and analyzed by TEM, which identified spherical material with a diameter of 150 nm. The concentration of the material with a diameter between 100 and 150 nm was also the highest in the Nanosight analysis (Figure 10A). Both TSG101 and CD63, as known EXO markers, were expressed (Figure 10B). EXOs were extracted from Ca9-22, OSC20, OSC19, and SAS cells upon addition of *P.g* LPS, and the expression level of PD-L1 was compared with that of EXOs from normal conditions without LPS. The level of PD-L1 expression in EXOs from OSCCs treated with LPS was higher (Figure 11). PD-L1 is released into the tumor microenvironment as EXOs.

## 4. Discussion

PD-L1, which is expressed on the surface of tumor cells, binds to PD-1 on the surface of cytotoxic T cells and suppresses T cell activity, which not only causes tumor immune escape but has also been implicated in tumor malignancy [33]. Patients with recurrence after initial cancer treatment who are not candidates for salvage surgery or radiotherapy, or who have distant metastases, are candidates for cancer drug therapy. Previously, 5-FU plus cisplatin/carboplatin (FP/FC) plus cetuximab therapy was the standard of care for platinum-sensitive recurrent or metastatic squamous cell carcinoma of the head and neck, based on the results of the EXTREME trial [34]. In March 2017, the use of nivolumab, an anti-PD-1 antibody, for patients with platinum-resistant recurrent or metastatic squamous cell carcinoma of the head and neck, was approved on the basis of the CheckMate 141 trial [19], and in December 2019, the use of PEM. An anti-PD-1, +FP/FC or PEM monotherapy in PD-L1-positive patients by combined positive score in those with platinum-sensitive recurrent or metastatic squamous cell carcinoma of the head and neck, was approved on the basis of the KETNOTE-048 trial [35]. 

Blockade of the PD-L1/PD-1 pathway has demonstrated remarkable antitumor effects in cancer patients and is recognized as the gold standard for developing new immune checkpoint blockade and combination therapies [36]. However, some tumors have limited response to anti-PD-L1 antibodies [37]. Therefore, understanding this PD-L1 expression mechanism will lead to further therapeutic advances for cancer patients.

TLR4 is activated by LPS, a component of Gram-negative bacteria, and induces the production of inflammatory mediators [38]. Oral squamous cell carcinomas have glycoproteins on their surfaces, which facilitate bacterial attachment. They are then exposed to oral commensal bacteria, forming a unique microflora called the microbiome. [39]. We hypothesized that this microbiome may cause various changes in oral cancer tissues and focused on *P.g*, which is a representative causative agent of periodontal disease and has recently been reported to be increased in microbiomes on tumor surfaces [40]. Periodontal disease is one of the most common bacterial infections, and bacterial epidemiology studies have shown a significant relationship between periodontal disease and oral cancer [41,42,43,44]. In addition, periodontal disease has recently been reported to be associated with mortality from oral gastrointestinal cancers, and *P.g*, a major periodontal pathogen, has been identified as a factor that increases the risk of death from oral gastrointestinal cancers [45]. We had already found in preliminary experiments that *P.g* activates the ERK1/2-ETS1, p38/HSP27, and PAR2/NFκB pathways to induce proMMP9 expression, followed by proenzyme activation by gingipain, to promote cell invasion into OSCC cell lines [46]. However, the details are not yet known regarding what induces the expression of MMP9 and promotes cell invasion. Therefore, we hypothesized that LPS of *P.g* might have a negative effect on oral cancer.

Our data showed that the addition of LPS from *P.g* and *E. coli* to OSCC cells did not alter cell proliferative capacity, migration ability, or cell morphology, but enhanced cell invasive capacity and PD-L1 expression and plasticity. Thus, it was suggested that the addition of bacterial LPS induces the expression of PD-L1, but does not change the nature of the cancer itself.

TGF-β is an important driver of EMT, regulating the transcription of downstream target genes and activating downstream signaling pathways to induce EMT [47]. We observed enhanced expression of the EMT–TF system and MMPs in OSCC with TGF-β by RT-PCR, and a similar response when *P.g* LPS was added, suggesting that EMT may be occurring even with the addition of *P.g* LPS.

EXOs are produced in the endosomal compartment of most eukaryotic cells and contain a wide range of functional proteins such as chemokines and cytokines, as well as mRNA and miRNA. EXO-mediated signaling is thought to be responsible for signaling between the tumor and surrounding stromal cells, activation of proliferative and angiogenic pathways, and formation of an environment suitable for metastasis, the pre-metastatic niche [48,49,50]. In the mechanism of tumor metastasis, it is becoming clear that the pre-metastatic niche has significantly higher levels of tumor cell-derived EXOs in the metastatic organs, and that these EXOs are taken up by future metastatic sites, creating an environment favorable for metastasis [51]. Yang et al. [52] successfully isolated EXOs from human breast cancer cells and mouse mammary tumor cells. They demonstrated that administration of PD-L1-positive EXOs promotes tumor growth in a mouse mammary tumor model. In addition, Chen et al. demonstrated enhanced secretion of PD-L1-positive EXOs mediated by IFN-γ signaling in a malignant melanoma model [53]. In these experiments, we confirmed the extraction of EXOs from OSCC cells and also PD-L1 expression within EXOs. Furthermore, the addition of LPS upregulated PD-L1 expression in EXOs, suggesting that EXOs may mediate the upregulation of PD-L1 expression after the addition of LPS, forming a pre-metastatic niche.

There are several limitations of this study. This was an in vitro study, so we should examine PD-L1 expression under LPS stimulation of *P.g* in 3D culture, which is closer to in vivo conditions, and/or in vivo by developing an oral cancer xenograft model using immunodeficient mice.

## 5. Conclusions

PD-L1 is involved in the invasion and metastasis of oral cancer via pEMT, and bacterial LPS as well as TGF-β was found to be involved in PD-L1 expression. In addition, OSCC EXOs supplemented with LPS highly expressed PD-L1, suggesting that EXOs may be involved as carriers of PD-L1.

## Figures and Tables

**Figure 1 cancers-16-00343-f001:**
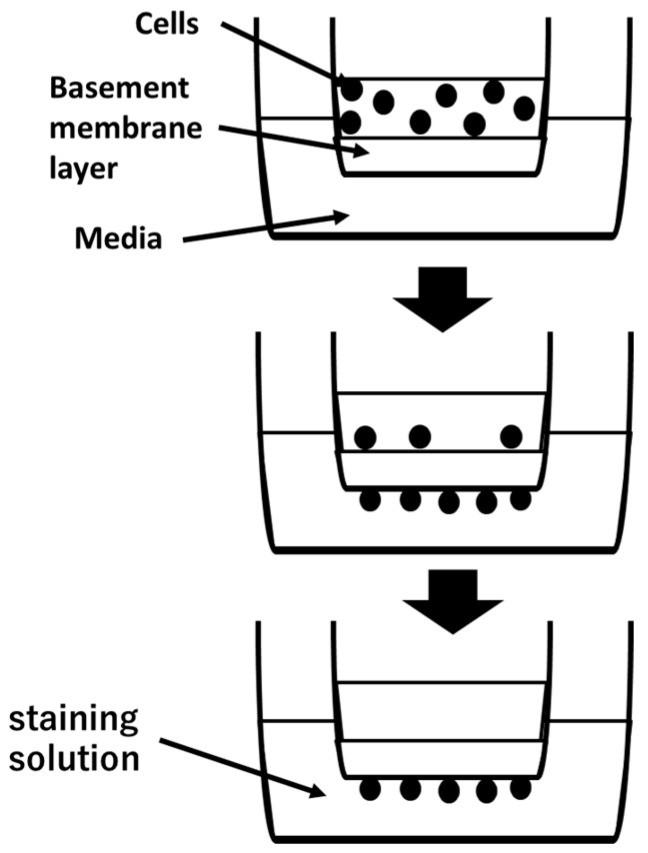
Cell suspension is placed in upper camber. Invasive cells pass through basement membrane layer and cling to the bottom of the insert membrane. After removal of non-invasive cells, invasive cells are stained and quantified.

**Figure 2 cancers-16-00343-f002:**
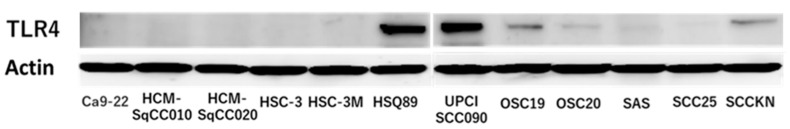
Expression of TLR4 in 12 OSCCs. TLR4 positivity in HSQ89, UPCISCC090, OSC19, OSC20, SAS, and SCCKN. The uncropped bolts are shown in Appendix A.

**Figure 3 cancers-16-00343-f003:**
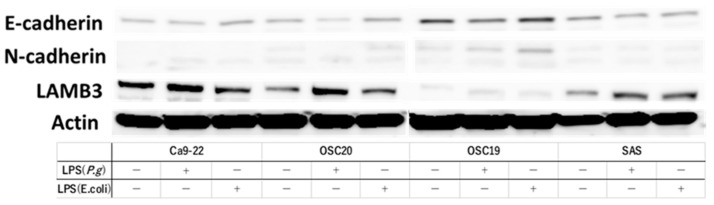
Evaluation of pEMT by expression of E-cadgerin, N-cadherin and expression of LAMB3, a marker of pMET, by addition of LPS. We used OSC19, OSC20, and SAS as TLR4-positive cells and Ca9-22 as TLR4-negative cells. We found an increase or decrease in E-cadherin and N-cadherin and an increase in LAMB3 in OSC19, OSC20, and SAS with the addition of LPS. The uncropped bolts are shown in Appendix A.

**Figure 4 cancers-16-00343-f004:**
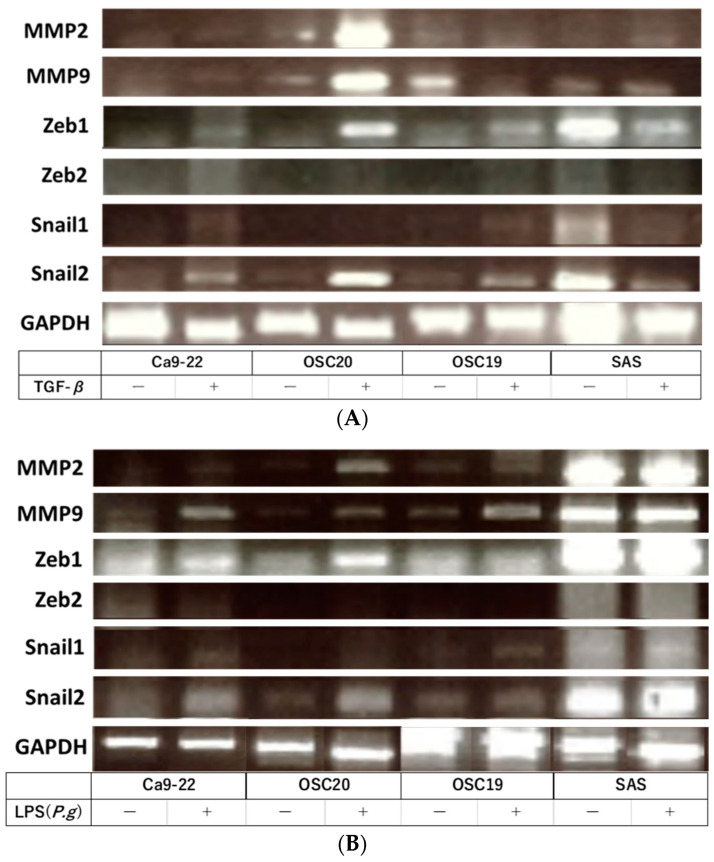
RT-PCR. The addition of TGF-β enhanced the expression of the gelatinase group of MMPs or EMT–TF system (**A**). Similar upregulation was also observed when *P.g* LPS was added (**B**). The uncropped bolts are shown in Appendix A.

**Figure 5 cancers-16-00343-f005:**
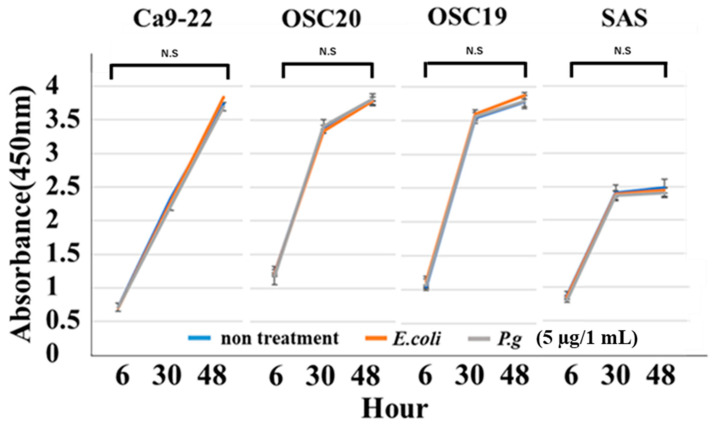
Cell proliferation ability with or without LPS addition. When *P.g* or *E.coli* LPS was added to OSCC cells, their proliferative abilities were unchanged. Bars denote the standard deviations (SDs). Compared to control culture and not significant (N.S), *p* ≥ 0.05.

**Figure 6 cancers-16-00343-f006:**
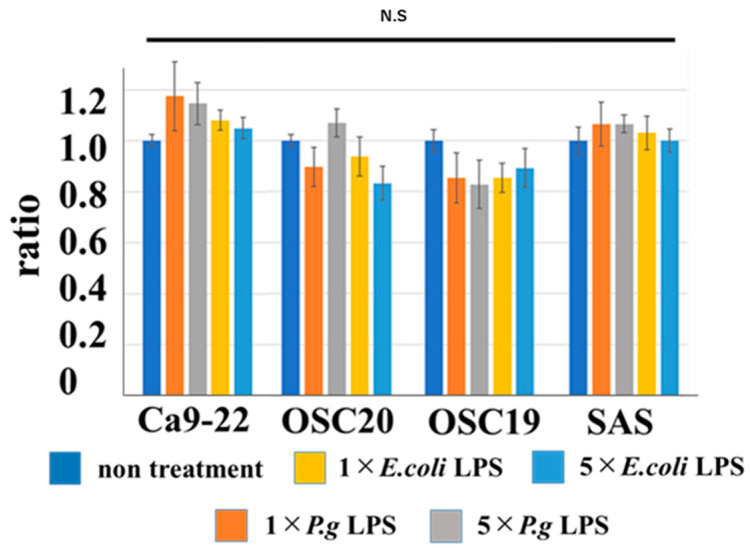
Cell migration ability with or without LPS addition. When *P.g* or *E.coli* LPS was added to OSCC cells, their migration abilities were unchanged. Bars denote the standard deviations (SDs). Compared to control culture and N.S, *p* ≥ 0.05.

**Figure 7 cancers-16-00343-f007:**
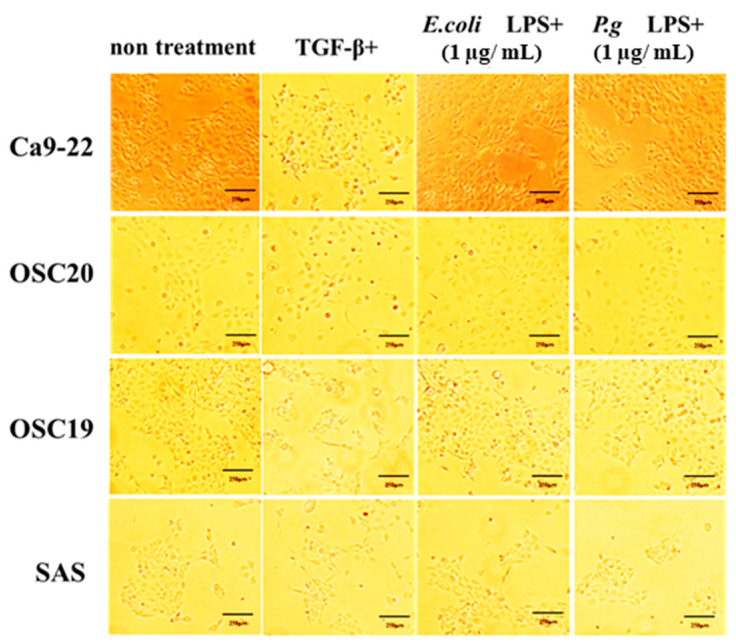
Changes in cell morphology with or without LPS addition. The spindle shape of the four OSCC cell lines was observed by adding TGF-β or *E. coli* LPS but not by the addition of *P.g* LPS. Scale bar, 250 μm.

**Figure 8 cancers-16-00343-f008:**
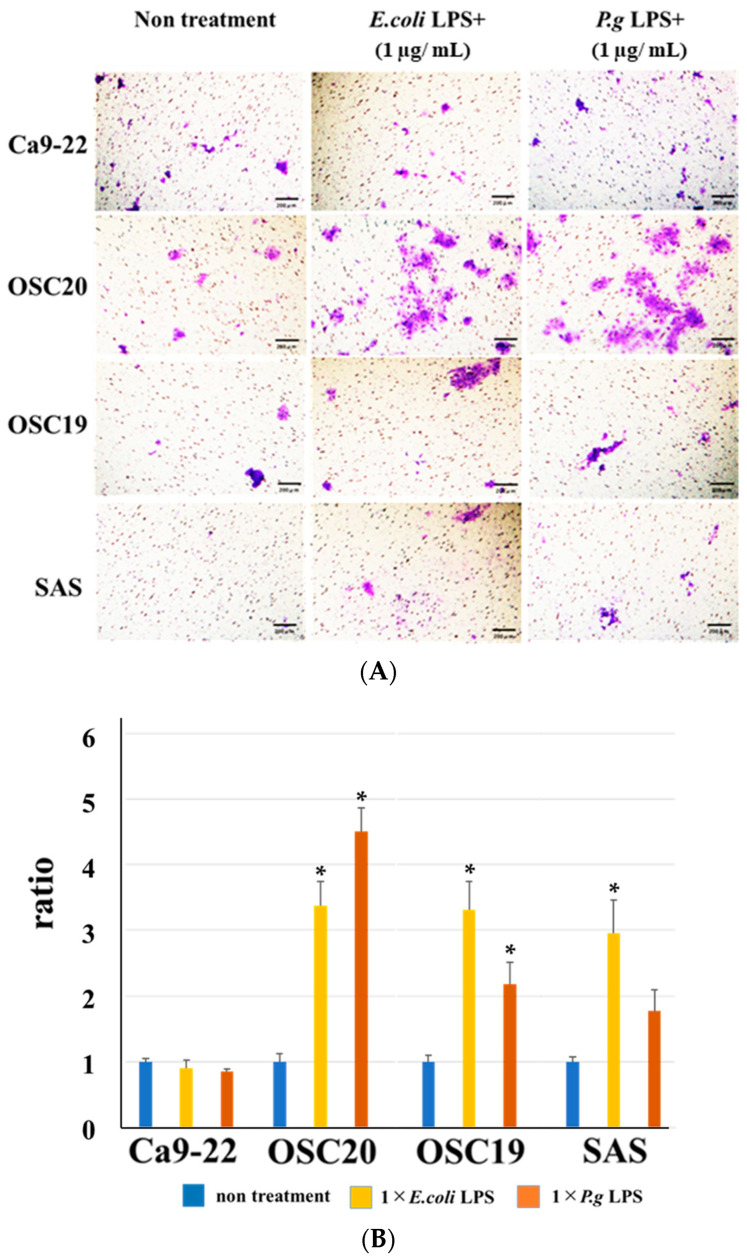
Cell invasion ability with or without LPS addition. The addition of *P.g* or *E. coli* LPS increased cell infiltration. Especially in OSC20, a strong increase in invasive capacity was observed. Scale bar, 200 μm (**A**). Bars denote the standard deviations (SDs). * *p* < 0.05 (**B**).

**Figure 9 cancers-16-00343-f009:**
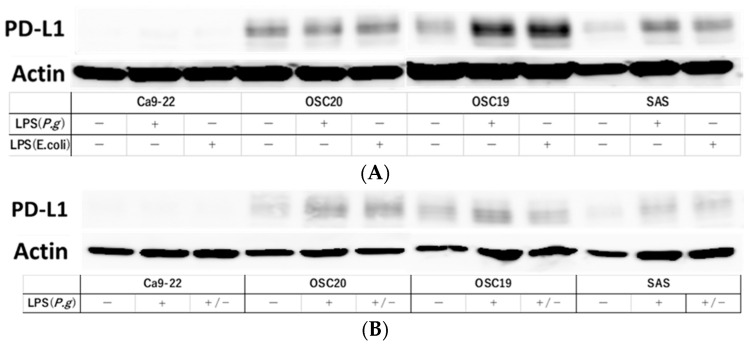
Evaluation of PD-L1 expression and reversibility of PD-L1 by addition of LPS. PD-L1 expression was increased by LPS stimulation in OSC20, OSC19 and SAS cells expressing TLR4 (**A**). Decreased expression of PD-L1 was observed in OSCC cells cultured in *P.g* LPS and then in LPS-free medium for 2 days (**B**). The uncropped bolts are shown in Appendix A.

**Figure 10 cancers-16-00343-f010:**
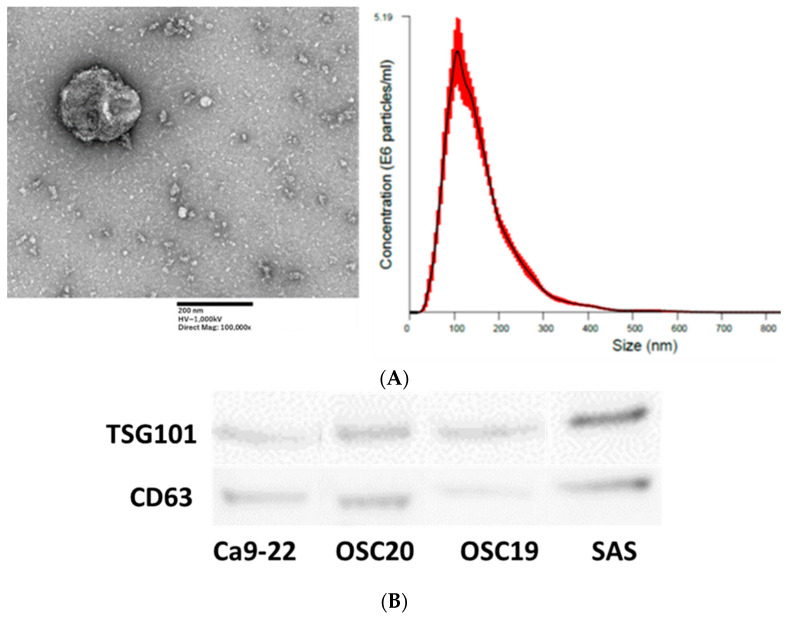
Nanosite analysis of EXOs by electron microscopy. Scale bar, 200 nm (**A**). EXO marker expression was confirmed in EXOs extracted from four OSCCs (**B**). The uncropped bolts are shown in Appendix A.

**Figure 11 cancers-16-00343-f011:**
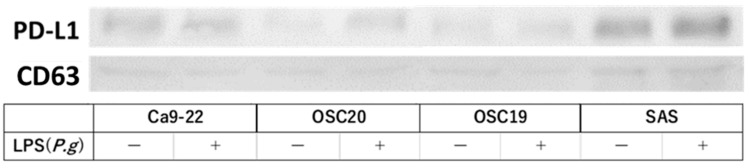
Addition of LPS enhanced PD-L1 expression in EXOs. The uncropped bolts are shown in Appendix A.

## Data Availability

Data are contained within the article.

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
