# Peer review of "Bacterial Lipopolysaccharide Induces PD-L1 Expression and an Invasive Phenotype of Oral Squamous Cell Carcinoma Cells"

_cancers, 2024, doi:10.3390/cancers16020343_

Round 1
Reviewer 1 Report
Comments and Suggestions for Authors
Need some editing to make reading more clear under abstract section in which you need to be more specific about findings ( just list them). Also the conclusion should read exactly as what you wrote under your discussion.
Towards the end of your manuscript, under discussion, you talked about EXOs, which seems to be an interesting finding to backed your hypothesis. However, you don’t refer to it in your abstract at all. Was that a last minute decision to add it. Reasons? Could you also explain why didn’t you do 3D as part of the original research project?
Moderate English Editing
Author Response
Reviewer 1
We are grateful to you for your time, effort, thoughtful suggestions.
We confirm that this manuscript has been improved by correcting the points you have indicated.
Indicating point
Need some editing to make reading more clear under abstract section in which you need to be more specific about findings ( just list them). Also the conclusion should read exactly as what you wrote under your discussion.
Towards the end of your manuscript, under discussion, you talked about EXOs, which seems to be an interesting finding to backed your hypothesis. However, you don’t refer to it in your abstract at all. Was that a last minute decision to add it. Reasons? Could you also explain why didn’t you do 3D as part of the original research project?
Response: Thank you for opinion. The Abstract has been added because the contents of EXO were not written in the Abstract.
The Conclusion has also been revised to include the EXO content and to be in line with the text.
Since 3D culture may cause pEMT due to a decrease in oxygen partial pressure in the center of the spheroid, and it may be difficult to determine whether it is due to the effect of LPS or not, only 2D was considered in this study.

Reviewer 2 Report
Comments and Suggestions for Authors
In this study, a battery of OSCC cell lines challenged with P. gingivalis LPS were assayed for expression of PD-L1 and for partial epithelial-mesenchymal transition. In general, the reported results were highly variable between cell lines in nearly all assays. Unfortunately, this is not well reflected in the text. The authors make numerous strong statements based only on results from one cell line that are clearly not seen in other cell lines.
Specific comments:
· Methods: invasion assay. Please include details (and references) for the invasion assay, including the specific type of plate/membrane/insert system used.
· Discussion, lines 305-306: "We hypothesized that this (tumor-associated) microbiome may cause epigenetic changes in oral cancer tissues..." This is an interesting hypothesis. The authors report no experiments testing it.
· Discussion, line 316: The reviewer presumes that "gingipain" is meant here. Note that "zingipain" is produced by ginger plants.
Author Response
Reviewer 2
We are grateful to you for your time, effort, thoughtful suggestions. I am confident that this manuscript has been improved by correcting the points you have pointed out.
Indicating point
In this study, a battery of OSCC cell lines challenged with P. gingivalis LPS were assayed for expression of PD-L1 and for partial epithelial-mesenchymal transition. In general, the reported results were highly variable between cell lines in nearly all assays. Unfortunately, this is not well reflected in the text. The authors make numerous strong statements based only on results from one cell line that are clearly not seen in other cell lines.
Response: Thank you for opinion. As you pointed out, differences in sensitivity to LPS were observed between the cell lines. This difference in susceptibility is likely to be causally related in some way, regardless of TLR-4 expression, primary site, degree of differentiation, or HPV infection. We believe that this phenomenon can be called p-EMT, since we were able to confirm a similar phenomenon with LPS as with the EMT generated by TGF-β, as well as the transcription factors expressed at the time of EMT induction. It is not possible to determine from this study whether this phenomenon is specific to certain types of cultured oral squamous cell carcinoma cells or general, but it can be said that it is not a general phenomenon but a phenomenon seen in specific cell lines, such as OSC-20. Although this study was not able to address the differences in susceptibility to LPS, we will continue to analyze cell lines using RNAseq to determine which characteristics of cell lines are induced by LPS to induce EMT.
Specific comments:
- Methods: invasion assay. Please include details (and references) for the invasion assay, including the specific type of plate/membrane/insert system used.
Response: In this study, the invasion assay was performed using the CytoSelectTM 24-well cell invasion assay kit according to the manufacturer's instructions (Cell Biolabs, San Diego, CA). The reference is Lee J, Kim JC, Lee SE, Quinley C, Kim H, Herdman S, Corr M, Raz E. Signal transducer and activator of transcription 3 (STAT3) protein suppresses adenoma-to-carcinoma transition in Apcmin/+ mice via regulation of Snail-1 (SNAI) protein stability. J Biol Chem. 2012 May 25;287(22):18182-9.
Discussion, lines 305-306: "We hypothesized that this (tumor-associated) microbiome may cause epigenetic changes in oral cancer tissues..." This is an interesting hypothesis. The authors report no experiments testing it.
Response: As you pointed out, we did not examine changes at the DNA level in this study, and we do not think that the term "epigenetic changes" is appropriate, so we will change the term to "various changes".
Discussion, line 316: The reviewer presumes that "gingipain" is meant here. Note that "zingipain" is produced by ginger plants.
Response: It is gingipain, not zingipain. Corrected.
Reviewer 3 Report
Comments and Suggestions for Authors
This manuscript is interesting in negative-result presentation which is fits to journal's aims and scope. However, the preparation is poor. Major modifications need to be done.

Author Response
Reviewer3
We are grateful to you for your time, effort, thoughtful suggestions.
We confirm that this manuscript has been improved by correcting the points you have indicated.
Indicating point 1
All titles including main and subtitles should be bold
Response: The main title and subtitle have been revised to bold.
Indicating point 2
Borders and sizes of images are not well arranged and aligned Figs 1, 2, 3, 4, 8, 9, and 10.
Response: Image borders and placement have been corrected.
Indicating point 3
Matrigel is used to mimic the basement membrane in vitro.
- Page 4, line 128-140, invasion assay, why Matrigel was not used in upper
chamber (interior insert)? Author can check Chea C et al 2023
(https://doi.org/10.3390/pharmaceutics15020562) for the information.
- Author should include Schematic representation of chamber migration and
invasion assays in this manuscript.
- Instead of scratching assay, author should perform chamber migration assay.
Alternatively, scratching-assay data should be placed in supplemental data.
Response: In this study, the invasion assay was performed using the CytoSelectTM 24-well cell invasion assay kit according to the manufacturer's instructions (Cell Biolabs, San Diego, CA). The kit uses a polycarbonate membrane with a pore size of 8 µm and is coated with a basement membrane matrix. We have added this information because it was omitted. We have also included a schematic diagram in the manuscript.
Indicating point 4
To all figure legends, more information about data curation should be added, eg., replication, scale bars, SD, SE, etc.
Response: We have added the necessary information to all Figure legends.
Indicating point 5
Table 1 and 2 should be placed into supplemental data
Response: Tables 1 and 2 have been added to the supplemental data.
Indicating point 6
Page 2 and 3, Briefly explained WB methodology in your manuscript.
Response: WB content of Materials and Methods has been added.
Indicating point 7
Fig 2, How do authors explain the relationship of E-cadherin to N-cadherin in highly E-cadherin expressing cells and lower ones?
Response: With the addition of LPS in P.g E-cad expression is decreased and N-cad expression is increased (especially in OSC19 and SAS), suggesting that this indicates a transition from epithelial to mesenchymal. Moreover, the increased expression of LAMB3 is considered to indicate that pEMT is occurring and that the condition has both mesenchymal and epithelial characteristics.
Indicating point 8
Fig 3, Western blotting with either RT-PCR or qPCR should be presented, and each result must be performed in the same gel.
- Separate/label the panel of TGF beta and P.g LPS stimulated results, e.g, Fig 3a, Fig 3b.
- Improve quality bands and running your RT-PCR . I am wonder that primers are specific and cycles are appropriate?
- Put frames surrounding each result and make a space from one to another.
Response: As you saw in the previously attached document, all WB data are using the same gel. We would appreciate it if you could double check.
Figure 3a shows TGF-β stimulation and Figure 3b shows P.g LPS stimulation.
RT-PCR is performed as in this cited paper (Noguchi K, Kanda S, Yoshida K, Funaoka Y, Yamanegi K, Yoshikawa K, Takaoka K, Kishimoto H, Nakano Y. Establishment of a patient-derived mucoepidermoid carcinoma cell line with the CRTC1-MAML2 fusion gene. Mol Clin Oncol. 2022, 3, 75).
We also framed the PCR results in Figure 3 and spaced them one by one.
Indicating point 9
Show statistical analysis to all figures necessary
Response: All figures have been statistically processed.
Indicating point 10
Fig 6, phase contrast microscope/image with higher magnification should be used to see clearly cell shape/transformation.
Response: I have processed the figure in high contrast
Indicating point 11
Fig 7, higher magnification should be used to see clearly invasive cells.
Response: I have processed the figure in high contrast
Indicating point 12
Scale bars need to be seen and details in figure legends in Fig 6, 7.
Response: Scale bar information has been added to Figure legend in Figures 6 and 7.
Indicating point 13
Erase x4 under fig 7 panel
Response: We have deleted the description of x4.
Indicating point 14
Intensity of all bands from WB or RT-PCR should be measured and included.
Response: The band ratios of WB are shown in the attached document, and RT-PCR is additionally described.
Indicating point 15
Fig 8, separate/label the panel of LPS stimulated results, e.g, Fig 8a, Fig 8b. Why P.g LPS induces similar expression or higher of PD-L1 than E.coli LPS?
Response: We think that P.g LPS increases PD-L1 expression via TLR4 expressed on tumor cells. In this study, we used E.coli LPS as a positive control. In this study, we were not able to determine the difference in sensitivity to LPS, but we will continue our analysis using RNAseq to determine what characteristics of cell lines are induced by LPS in EMT.
Indicating point 16
Fig 9, was that result of TSG101 running at the same gel?
Response: This is the result of a reaction with the same gel.
Indicating point 17
Author should explain why proliferation and migration are not affected by P.g or E.coli LPS? Transcriptional factors Snail1, Snal2, Zeb1, and Zeb2 must be explained and discussed more in this manuscript.
Response: The addition of LPS did not increase proliferation or migration capacity, but it did increase invasive capacity. This indicates that the addition of LPS increased invasive ability by rotating the EMT cycle, but did not affect the cell cycle of the cancer cells, which is considered to indicate that there was no change in proliferative and migratory ability.
Indicating point 18
Why TSG101, CD63, and MMP2 are presented in this article?
Response: TSG101 and CD63 were examined as they are markers for EXOs. MMP2.9 was also examined as a cause of ECM degradation by Invasion assay.
Indicating point 19
What is/are novel in this article?
Response: The novelty of this paper is that factors related to malignancy, such as p-EMT and PD-L1, are altered by external factors other than the microenvironment in contact with cancer cells.
Although there have been papers showing that oral bacteria have negative effects on other organs, there have been no reports of oral bacteria increasing invasiveness and altering p-EMT and PD-L1 in oral cancer, which is highly novel.
Round 2
Reviewer 2 Report
Comments and Suggestions for Authors
The authors have adequately addressed most of this reviewer's concerns.
However, the revised Conclusion in the Abstract goes well beyond what the data support. The Conclusion in Version 1 was more appropriate.
Author Response
Reviewer 2
We are grateful to you for your time, effort, thoughtful suggestions.
We confirm that this manuscript has been improved by correcting the points you have indicated.
Indicating point
The authors have adequately addressed most of this reviewer's concerns.
However, the revised Conclusion in the Abstract goes well beyond what the data support. The Conclusion in Version 1 was more appropriate.
Response: Thank you for opinion.
This time, Reviewer1 pointed out that the conclusion of the Abstract did not include the contents of EXOs, so we have revised the Abstract. As you pointed out, I thought that the conclusion to the results in the Abstract was a bit of a leap, so I have revised it.